# Relationship between Food Dependence and Nicotine Dependence in Smokers: A Cross-Sectional Study of Staff and Students at Medical Colleges

**DOI:** 10.3390/medicina55050202

**Published:** 2019-05-23

**Authors:** Yutaka Owari, Nobuyuki Miyatake, Hiromi Suzuki

**Affiliations:** 1Shikoku Medical College, Utazu, Kagawa 769-0205, Japan; 2Department of Hygiene, Faculty of Medicine, Kagawa University, Miki, Kagawa 761-0793, Japan; miyarin@med.kagawa-u.ac.jp (N.M.); tanzukimama@yahoo.co.jp (H.S.)

**Keywords:** food dependence, interdependence, nicotine dependence

## Abstract

*Background and objectives:* The aim of this study was to examine the relationship between nicotine dependence and food dependence in smokers. Smoking and obesity are both serious public health problems that give rise to diseases and increased medical expenses. Nicotine dependence is one of the sources of difficulty in smoking cessation, while food dependence is one of the causes of obesity. *Materials and Methods:* We examined the data of 72 (smoking vs. nonsmoking) and 62 (nicotine dependence vs. no nicotine dependence) subjects among 321 staff and students at medical colleges in Kagawa and Okayama prefectures in Japan. *Results:* There was a significant difference in food dependence (except women) between the smoking and nonsmoking groups (total: smoking 4.7 ± 6.1, nonsmoking 2.1 ± 2.0, *p* = 0.0411; men: smoking 4.0 ± 4.7, nonsmoking 2.0 ± 2.1, *p* = 0.0490). There was also a significant difference in food dependence (except women) between the nicotine dependence and no nicotine dependence groups (total: nicotine dependence 4.6 ± 6.3, no nicotine dependence 2.0 ± 2.1, *p* = 0.0370; men: nicotine dependence 3.6 ± 4.8, no nicotine dependence 1.6 ± 1.8, *p* = 0.0489). *Conclusion:* The findings showed that the smoking group (and nicotine dependence group) had higher food dependence than the nonsmoking group (and no nicotine dependence group). Our results indicate an interdependence between nicotine and food dependences.

## 1. Introduction

Smoking and obesity are both serious public health problems that give rise to diseases and increased medical expenses. In Japan, 1.03 million people were estimated to have smoking-related diseases and annual related medical expenses were estimated to be 1.5 trillion yen in 2017 [1]. Various approaches that promote smoking cessation have been explored, however, they have yet to prove sufficiently effective. Nicotine dependence is one of the sources of difficulty in smoking cessation [2]. In addition, obesity is closely associated with health problems and increased medical expenses [3]. Food dependence, the symptom count of food addiction as a food dependence score, is also considered to be one of the causes of obesity [4].

The time discount rate is a ratio that shows how much discount is considered when comparing current satisfaction with future satisfaction [5]. Normally, the time discount rate is measured as the proportion of satisfaction that can be postponed to the future when receiving a fixed amount of money. In other words, the time discount rate is the degree of emphasis on current satisfaction versus future satisfaction. Many studies have reported that smokers had higher time discount rates than nonsmokers [6,7]. Furthermore, subjects with higher nicotine intake among smokers have higher time discount rates [8]. Similarly, obese people were shown to have higher time discount rates than non-obese people [9]. Therefore, nicotine and food dependences are thought to have the same structure as substance dependence [10], underlying the time discount theory. In addition, in nicotine and food dependences, their external stimuli induce a desire for substances through similar brain area activation. Food and smoking cues were both associated with increased blood oxygenation-dependent responses in the left amygdala, bilateral orbital cortex, and striatum. Brain regions identified here are involved in learning, memory, and motivation, and their healing induction activity is an indicator of the simulative saliency of cues [11]. However, few studies have shown a link between nicotine and food dependences by taking into account the time discount rate.

In this study, we investigated the link between nicotine and food dependences, as an interrelationship of two dependences.

## 2. Materials and Methods

### 2.1. Study Design

We conducted an observational cross-sectional study. Our objectives were to explore two hypotheses. First, the smoking group had a higher food dependence score than the nonsmoking group and the nicotine dependence group, especially, had a higher food dependence score than the no nicotine dependence group. Second, the smoking group had a higher time discount rate than the nonsmoking group and the nicotine dependence group, especially, had a higher time discount rate than the no nicotine dependence group.

### 2.2. Subjects

Between 1 September and 31 October 2018, we enrolled 321 (149 men, 172 women) staff and students of A and B medical colleges (in Kagawa and Okayama prefectures, Japan) in the study (Figure 1). The subjects completed self-administered questionnaire surveys twice. First, all subjects responded to the following: sex; age (years); height (cm); weight (kg); smoking history; annual income (less than 2 million yen, 2 million yen to less than 6 million yen, 6 million yen or over); academic background (high school graduate, vocational school/university graduate); and willingness to participate in the second survey. Many studies have shown that smoking varies depending on socioeconomic factors such as sex, age, income, and educational background [10,12,13,14], and therefore it was necessary to adjust for these confounding factors. We obtained data on 38 smokers (24 men, 16.1%; 14 women, 8.1%) and 283 nonsmokers. In the second survey, the subjects completed a food-dependent questionnaire survey. We were unable to obtain data on 2 of the smokers and 6 of the nonsmokers. We randomly extracted 36 people from 277 nonsmokers that corresponded to the following smokers’ conditions: sex, age (20–39, 40–59, 60 years old and over), annual income, and educational background. Therefore, the data used for comparison was 72 (smoking vs. nonsmoking) subjects. In addition, 31 of them (19 men and 12 women) were defined as being nicotine dependent (nicotine dependence group), and we selected a further 31 nonsmoking control subjects (no nicotine dependence group) from among the nonsmoking group. Therefore, the data used for comparison was 62 (nicotine dependence vs. no nicotine dependence). We received ethical approval from the Shikoku Medical College Ethics Screening Committee (approval number: H30-5; 29 April 2018). Each participant provided written informed consent.

### 2.3. Clinical Parameters and Measurements

Anthropometric and body composition parameters were evaluated as confounding factors based on the subjects’ age (years) and body mass index (BMI) (kg/m^2^).

### 2.4. Smoking Habits

Smoking habits were evaluated by the subjects’ response to the following question: Do you currently smoke tobacco products? (1) I am not currently smoking and have never smoked; (2) I smoked in the past, but do not currently smoke (more than 5 years after quitting smoking); (3) I currently smoke. Those who chose answer three were defined as subjects with (current) smoking, while those who answered one and two were defined as nonsmoking subjects. The risk of lifestyle-related diseases is greatly reduced by smoking cessation periods of 5 years or more [15,16]. There were no people within 5 years of smoking cessation.

### 2.5. Nicotine Dependence

Nicotine dependence was evaluated using the Tobacco Dependence Screener (TDS) [17] and the Brinkman index [18]. The Brinkman index is calculated as the average number of cigarettes per day (number) multiplied by the number of years of smoking. We defined nicotine dependence as more than five items in the TDS test, and the Brinkman index of more than 200. However, we used only the TDS test because the Brinkman index was inappropriate for those under 35 years of age.

### 2.6. Food Dependence

Food dependence was evaluated by using the 9-item short type Yale Food Addiction Scale (mYFAS). The original Yale Food Addiction Scale (YFAS) was established with 25 items. YFAS was the full version of mYFAS. The mYFAS was proposed in 2014 and it can be evaluated in the same way as the YFAS [19,20]. Although the YFAS has diagnostic criteria and symptom count criteria, we used the symptom count as the food dependence score according to the previous report in this study [21,22,23,24]. We used Ohnishi for the Japanese translation [25].

### 2.7. Time Discount Rate

We measured the time discount rate from the response to the following question, “If you are asked to wait a year to receive one million yen, how much interest would you want?” and the rate reported by the respondent was taken as the time discount rate [26]. A high time discount rate indicated a tendency to prioritize currently available utility rather than future utility.

### 2.8. Statistical Analyses

Continuous variables were presented as the mean ± standard deviation (SD). We used the Welch’s t-test and the Mann–Whitney U test to compare the averages of continuous variables between the groups (smoking vs. nonsmoking or nicotine dependence vs. no nicotine dependence). The threshold for significance was *p* < 0.05 and we performed a robust test called the bootstrap method to compensate for the shortcoming of small sample numbers [27]. All calculations were performed using STATA, version 14 (STATA Corp LLC., College Station, TX, USA).

## 3. Results

We used the data from 72 (smoking and nonsmoking groups) or 62 subjects (nicotine dependence and no nicotine dependence groups) from among the 321 respondents (Figure 1). As shown in Table 1 and Table 2, each group was homogenous after stratified matching.

First, there was a significant difference in food dependence (except women) between the smoking group and the nonsmoking group (total: smoking 4.7 ± 6.1, nonsmoking 2.1 ± 2.0, *p* = 0.0411; men: smoking 4.0 ± 4.7, nonsmoking 2.0 ± 2.1, *p* = 0.049) (Table 1). There was also a significant difference in food dependence (except women) between the nicotine dependence group and the no nicotine dependence group (total: nicotine dependence 4.6 ± 6.3, no nicotine dependence 2.0 ± 2.1, *p* = 0.037; men: nicotine dependence 3.6 ± 4.8, no nicotine dependence 1.6 ± 1.8, *p* = 0.0489) (Table 2).

Second, as shown in Table 1, there were significant differences in time discount rate (%) between the smoking and nonsmoking groups (total: smoking 11.3 ± 19.7, nonsmoking 2.4 ± 2.4, *p* = 0.0107; men: smoking 9.0 ± 14.3, nonsmoking 2.9 ± 2.6, *p* = 0.0109; women: smoking 15.9 ± 27.8, nonsmoking 1.4 ± 1.6, *p* = 0.0427). There were also significant differences in time discount rate (%) between the nicotine dependence and no nicotine dependence groups (total: nicotine dependence 12.3 ± 2.1, no nicotine dependence 2.3 ± 2.1, *p* = 0.0133; men: nicotine dependence 9.9 ± 15.8, no nicotine dependence 2.8 ± 2.3, *p* = 0.0434; women: nicotine dependence 15.9 ± 27.8, no nicotine dependence 1.4 ± 1.6, *p* = 0.0427) (Table 2). After carrying out bootstrap approval (10,000 samples), 0 was not included in the 95% confidence interval (CI) for all tests.

## 4. Discussion

First, the smoking group (and nicotine dependence group) had a higher food dependence score and higher time discount rate than the nonsmoking group (and no nicotine dependence group), validating our hypotheses. Our results are not consistent with many previous studies, however, some are consistent [28,29]. However, there was no significant difference for women with regard to food dependence, which seemed to be due to the small sample size (n = 12).

The nicotine dependence group was highly food dependent. We speculated that there was interdependence among these dependences. Therefore, in order to overcome dependence, resolving only one dependence is insufficient and it is necessary to resolve both dependences at the same time.

Second, the smoking group (and nicotine dependence group) had a higher time discount rate than the nonsmoking group (and no nicotine dependence group). This finding is consistent with many previous studies [6,7]. However, few other studies have reported that smokers (and nicotine-dependent subjects) simultaneously had a higher time discount rate and higher food dependence than nonsmokers (and no nicotine dependence subjects). Recently, it has been clarified in the field of neuroeconomics that the time discount rate is greatly influenced by substances in the brain such as serotonin [24,25,26]. Such influences may play critical roles in both nicotine and food dependences.

Eliza et al. [30] reported that the diagnostic construct of "food addiction" is a highly controversial subject. Overall, their findings support that food addiction is unique and consistent with criteria for other substance use disorder diagnoses. The evidence further suggests that certain foods, particularly processed foods with added sweeteners and fats, demonstrate the greatest addictive potential. Although, both behavioral and substance-related factors are implicated in the addictive process, the symptoms appear to better fit criteria for substance use disorder than behavioral addiction. We also carried out this study according to this view.

Several limitations of this study should be acknowledged. First, we could not secure a sufficient number of subjects. Our study reported that the smokers’ score was 6.2 ± 8.1 and the nonsmokers’ score was 2.3 ± 2.4 in food dependence (also nicotine-dependent score and no nicotine dependence) in women. The differences between the two groups were greater than that of men, however, there were no statistically significant differences in these groups. The reason for this seems to be that the sample size was small. Therefore, we should be cautious about generalizing the conclusions of this study. Secondly, we could not clarify the causal relationship among smoking dependence (and nicotine dependence), food dependence, and the time discount rate. We need to conduct a longitudinal study to clarify these causal relationships. Finally, the mYFAS has been verified to be as reliable as the YFAS. Thus, the mYFAS has reduced the burden on participants by reducing the overall number of questions, enabling the assessment of food poisoning in large epidemiological studies [31,32]. In this study, we used the mYFAS to clarify the relationship between food and nicotine dependences, and in the future, we would like to confirm the relationship between them using the full version of YFAS. Nevertheless, through strategies for resolving two dependences at the same time, we may be able to reduce the incidence of diseases and their related medical expenses.

## 5. Conclusions

The findings showed that the smoking group (and nicotine dependence group) had higher food dependence than the nonsmoking group (and no nicotine dependence group). Our results indicate an interdependence between nicotine and food dependences.

## Figures and Tables

**Figure 1 medicina-55-00202-f001:**
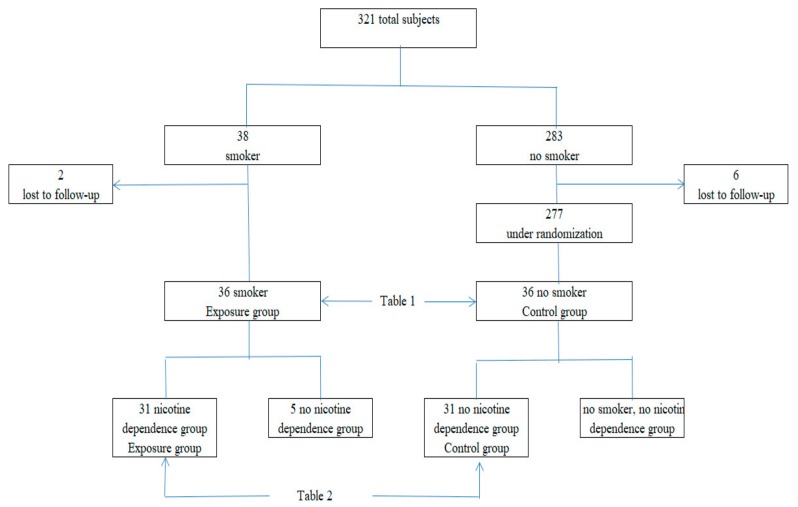
Screening, follow-up, and analysis of the study subjects.

**Table 1 medicina-55-00202-t001:** Clinical characteristics (smoking vs. nonsmoking).

**Total**	Smoking			Nonsmoking			Welch’s t-test	
Number of subjects	36			36				
Age (years)	41.7	±	17.7	40.4	±	20.0		
Body mass index (BMI) (kg/m^2^)	22.2	±	3.0	22.4	±	3.7		
Food dependence	4.7	±	6.1	2.1	±	2.0	**0.0411**	
Time discount rate (%)	11.3	±	19.7	2.4	±	2.4	**0.0107**	
Brinkman index	334.3	±	323.4					
Tobacco dependence screener (TDS)	5.9	±	2.6					
Nicotine dependence (presence)	31							
Income (/year) (number)	1:15, 2:19, 3:2							
Educational background (number)	1:26, 2:10							
**Men**	Smoking			Nonsmoking			Welch’s t-test	
Number of subjects	24			24				
Age (years)	37.4	±	16.6	36.3	±	19.1		
Body mass index (BMI) (kg/m^2^)	22.5	±	3.1	22.4	±	3.7		
Food dependence	4.0	±	4.7	2.0	±	2.1	**0.0490**	
Time discount rate (%)	9.0	±	14.3	2.9	±	2.6	**0.0107**	
Brinkman index	327.9	±	336.2					
Tobacco dependence screener (TDS)	5.9	±	2.9					
Nicotine dependence (presence)	19							
Income (/year) (number)	1:11, 2:12, 3:1							
Education background (number)	1:18, 2:6							
**Women**	Smoking			Nonsmoking			Mann–Whitney U test	
Number of subjects	12			12				
Age (years)	49.3	±	17.6	48.8	±	19.1		
Body mass index (BMI) (kg/m^2^)	21.7	±	3.1	22.4	±	3.7		
Food dependence	6.2	±	8.1	2.3	±	2.4	**0.3618**	
Time discount rate (%)	15.9	±	27.8	1.4	±	1.6	**0.0427**	
Brinkman index	347.3	±	310.2					
Tobacco dependence screener (TDS)	5.9	±	1.9					
Nicotine dependence (presence)	12							
Income (/year) (number)	1:4, 2:7, 3:1							
Educational background (number)	1:8, 2:4							

Income: 1, 200 less than one million; 2, 2 million or more but less than 6 million yen; 3, 6 million yen or more.

Educational background: 1, high school graduate; 2, vocational school/university graduate.

**Table 2 medicina-55-00202-t002:** Clinical characteristics (Nicotine dependence vs. No nicotine dependence).

**Total**	Nicotine dependence			No nicotine dependence			Welch’s t-test	
Number of subjects	31			31				
Age (years)	44.1	±	17.4	43.4	±	20.3		
body mass index (BMI) (kg/m^2^)	22.3	±	3.1	22.6	±	3.8		
Food dependency tendency	4.6	±	6.3	2.0	±	2.1	**0.0370**	
Time discount rate (%)	12.3	±	2.1	2.3	±	2.1	**0.0133**	
Brinkman index	377.0	±	328.2					
Tobacco dependence screener (TDS)	6.4	±	2.4					
Nicotine dependence (presence)	31							
Income (/year) (number)	1: 11, 2: 18, 3: 2							
Educational background (number)	1: 21, 2: 10							
**Men**	Nicotine dependence			No nicotine dependence			Mann–Whitney U test	
Number of subjects	19			19				
Age (years)	40.8	±	17.0	39.5	±	20.1		
body mass index (BMI) (kg/m^2^)	22.6	±	3.2	22.7	±	4.0		
Food dependence	3.6	±	4.8	1.6	±	1.8	**0.0489**	
Time discount rate (%)	9.9	±	15.8	2.8	±	2.3	**0.0434**	
Brinkman index	395.8	±	346.2					
Tobacco dependence screener (TDS)	6.6	±	2.7					
Nicotine dependence (presence)	19							
Income (/year) (number)	1: 7, 2: 11, 3:1							
Education background (number)	1: 13, 2: 6							
**Women (smoking = nicotine dependence)**	Nicotine dependence			No nicotine dependence			Mann–Whitney U test	
Number of subjects	12			12				
Age (years)	49.3	±	17.6	48.8	±	19.1		
body mass index (BMI) (kg/m^2^)	21.7	±	3.1	22.4	±	3.7		
Food dependence	6.2	±	8.1	2.3	±	2.4	**0.3618**	
Time discount rate (%)	15.9	±	27.8	1.4	±	1.6	**0.0427**	
Brinkman index	347.3	±	310.2					
Tobacco dependence screener (TDS)	5.9	±	1.9					
Nicotine dependence (presence)	12							
Income (/year) (number)	1: 4, 2: 7, 3: 1							
Education background (number)	1: 8, 2: 4							

Income: 1, 200 less than one million; 2, 2 million or more but less than 6 million yen; 3, 6 million yen or more.

Educational background: 1, high school graduate; 2, vocational school/university graduate.

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
