# Peer review of "Relationship between Food Dependence and Nicotine Dependence in Smokers: A Cross-Sectional Study of Staff and Students at Medical Colleges"

_medicina, 2019, doi:10.3390/medicina55050202_

Round 1
Reviewer 1 Report
Owari et al. present a study examining the relationship between nicotine dependence and food dependence in smokers. There is some novelty in the work as there is a scarcity of reports on the link between the two dependences that take into account the time discount rate, a ratio that indicates how much discount is considered when comparing current satisfaction with future satisfaction. The authors found there is a significant difference in food dependence between smoking and non-smoking groups as well as between nicotine-dependent and non-nicotine-dependent groups. They conclude that the smoking and nicotine-dependent groups have a higher food dependency than the non-smoking and non-nicotine-dependent groups, respectively. They interpret these results as indicating interdependence between nicotine dependence and food dependence.
Unfortunately, while the study addresses an interesting subject, it is of such a preliminary stage, as the authors admit, that reaching sound conclusions is difficult. I have a number of concerns, some major and some relatively minor.
1. The premise for the study needs to be better described. What is meant by the term, “structure” on line 45? This is an important concept underlying the basis for the work and needs to be fully explained.
2. The number of subjects actually used to generate data is confusing. The authors indicate that they recruited 321 people for the study (lines 16 and 59), yet actually utilized only 134 (line 15), if I understand correctly. The figure does not help clarify the situation.
3. The criteria used to classify subjects is questionable. First, one of the challenges with this type of study is that it is based on self-reporting. For example, three questions related to smoking habits were asked of potential subjects. There is no corroboration of subject answers. In addition, to include people who responded as former smokers in the non-smoking category is a mistake given the potential impact of prior smoking habits on food dependence (do they eat more to stay off of nicotine, for example?). They should be analyzed as a separate group.
4. The phrase “except women” is used throughout the manuscript. This should be revised as what it means is not clear in the context in which it is used.
5. A major issue regards the use of the mYFAS versus the full YFAS to determine food dependency. This is particularly puzzling in that the authors acknowledge this shortcoming (lines 157-158).
6. Another major concern is the sample size. The authors acknowledge this issue twice. First with respect to the lack of a significant finding using the data generated from females (line 141) and second, with the study as a whole, again as the authors acknowledge (lines 153-154). In addition, with respect to the female data, it is not clear how the authors surmise that significance was not found because of low sample size (line 141). Was power analysis done for any aspect of the study?
Author Response
1. The premise for the study needs to be better described. What is meant by the term, “structure” on line 45? This is an important concept underlying the basis for the work and needs to be fully explained.
(Answer) It is as per the teaching. Therefore, I added the following statement to lines 45-47 about the similarity between food dependence and nicotine dependence.
Lines 45-47.
Also, in nicotine dependence and food dependence, their external stimuli induce a desire for substances through similar brain area activation [11].
2. The number of subjects actually used to generate data is confusing. The authors indicate that they recruited 321 people for the study (lines 16 and 59), yet actually utilized only 134 (line 15), if I understand correctly. The figure does not help clarify the situation.
(Answer) It is as per the teaching. The supplementary explanation is as follows:
Of the 321 people who participated, there were 283 non-smokers. Of these, 277 people who obtained cooperation were classified into 36 classes according to gender, age, income, and educational background. Similarly, 36 of the 38 smokers who obtained cooperation were classified into 36 levels. After that, 36 people belonging to the same hierarchy as smokers were randomly extracted from non-smokers.
Figure:
This was inserted by the arrow in the figure.
3. The criteria used to classify subjects is questionable. First, one of the challenges with this type of study is that it is based on self-reporting. For example, three questions related to smoking habits were asked of potential subjects. There is no corroboration of subject answers. In addition, to include people who responded as former smokers in the non-smoking category is a mistake given the potential impact of prior smoking habits on food dependence (do they eat more to stay off of nicotine, for example?). They should be analyzed as a separate group.
(Answer) It is as per the teaching. In order to make a more objective assessment, in addition to smokers' self-reports, we should have conducted a breath carbon monoxide concentration measurement and a urine nicotine check test, but only self-reported data could be obtained.
As there was a drop in part of the additional materials, we have added this time.
Line 85.
(more than 5 years after quitting smoking)
Lines 87-88.
The risk of lifestyle-related diseases is greatly reduced by smoking cessation periods of 5 years or more [15,16]. There were no people within 5 years of smoking cessation.
4. The phrase “except women” is used throughout the manuscript. This should be revised as what it means is not clear in the context in which it is used.
(Answer) It is as per the teaching. Our study reported that smokers’ score was 6.2 ± 8.1 and non-smokers’ score was 2.3 ± 2.4 in food dependence (also nicotine-dependent’ score and non-nicotine-dependence’) in women. The difference between the two groups is greater than that of men. But, there were not statistically significant differences in these groups. The reason seems to be that the sample size was small.
5. A major issue regards the use of the mYFAS versus the full YFAS to determine food dependency. This is particularly puzzling in that the authors acknowledge this shortcoming (lines 157-158).
(Answer) It is as per the teaching. Therefore, I changed lines 161-163 as follows.
Lines 169-171.
In this study, we could use mYFAS to clarify the relationship between food dependence and nicotine dependence, but in the future, we would like to confirm the relationship between them using full version of YFAS.
6. Another major concern is the sample size. The authors acknowledge this issue twice. First with respect to the lack of a significant finding using the data generated from females (line 141) and second, with the study as a whole, again as the authors acknowledge (lines 153-154). In addition, with respect to the female data, it is not clear how the authors surmise that significance was not found because of low sample size (line 141). Was power analysis done for any aspect of the study?
(Answer) It is as per the teaching. Therefore, I performed a robust test called the bootstrap method to compensate for the shortcoming of small sample numbers. In addition, the subtitle "Pilot research" was added to the title.
2. Materials and Methods
2.8. Statistical analysis
Line 112-113.
And, we performed a robust test called the bootstrap method to compensate for the shortcoming of small sample numbers [Efron, B. Bootstrap methods: another look at the jackknife. Ann Stat. 1979, 7, 1-26].
3. Results
Line 133-134.
After carrying out bootstrap approval (10,000 samples), 0 was not included in the 95% confidence interval (CI) for all test.

Reviewer 2 Report
This is a innovative and well-written study. Despite some sample size limitations, the findings are provocative and intriguing.
The authors should address the following points that would add clarity to the findings.
Nicotine is an appetite suppressant so it may seem a little surprising that highly dependent smokers are more food-dependent (at least in men).
In contrast, heavy smokers may also be more food dependent. There is some literature on this, and this might explain the findings.
The concept of food dependency should be refined. It is likely that cravings for food are greater for specific foods rather e.g. sugar.
Author Response
This is a innovative and well-written study. Despite some sample size limitations, the findings are provocative and intriguing.
The authors should address the following points that would add clarity to the findings.
Nicotine is an appetite suppressant so it may seem a little surprising that highly dependent smokers are more food-dependent (at least in men).
In contrast, heavy smokers may also be more food dependent. There is some literature on this, and this might explain the findings.
The concept of food dependency should be refined. It is likely that cravings for food are greater for specific foods rather e.g. sugar.
(Answer) It is as per the teaching. The following two points have been added to the discussion section regarding the two points pointed out.
1. “Nicotine is an appetite suppressant so it may seem a little surprising that highly dependent smokers are more food-dependent (at least in men).
In contrast, heavy smokers may also be more food dependent. There is some literature on this, and this might explain the findings.”
Discussion
Lines 144-145.
Our results are not consistent with many previous studies, but some are consistent (Arnaud et al., 2008; Ariana et al., 2016).
2. The concept of food dependency should be refined. It is likely that cravings for food are greater for specific foods rather e.g. sugar.
Discussion
Lines 158-164.
Eliza et al., reported that “The diagnostic construct of "food addiction" is highly controversial subject.・・・Overall, findings support food addiction as a unique consistent with criteria for other substance use disorder diagnoses. The evidence further suggests that certain foods, particularly processed foods with added sweeteners and fats, demonstrate the greatest addictive potential. Though both behavioral and substance related factors are implicated in the addictive process, symptoms appear to better fit criteria for substance use disorder than behavioral addiction.” We also carried out this study according to this view.

Round 2
Reviewer 1 Report
1. The authors still do not explain what is meant by the term, “structure” on line 45. While they have added a more useful sentence on lines 45-47, they should expand on it. To what brain areas are they referring?
2. I do not understand what, “It is as per the teaching”, means. This is most likely a language issue but not one I can figure out. Regardless, using the identical phrase throughout the authors’ response is not appropriate.
3. The confusion concerning the number of subjects remains (no changes to the text, addition of two arrows to the figure). They state in the abstract (line 15) that data from 134 people were examined, yet they state that 321 people were recruited for the study (lines 16 and 59). The authors’ response adds to the confusion as now they are talking about 36 classes and 36 levels.
4. The authors have adequately addressed my concern regarding the criteria used to classify subjects.
5. While the authors have addressed my issue with the phrase “except women” in their response, they have not modified the text to explain the phrase.
6. The authors do not adequately address the major issue of using the mYFAS versus the full YFAS to determine food dependency other than to say they “would like” to use it in the future.
7. The authors have adequately addressed my concern regarding statistical analysis (bootstrap).
Author Response
Comments and Suggestions for Authors
1. The authors still do not explain what is meant by the term, “structure” on line 45. While they have added a more useful sentence on lines 45-47, they should expand on it. To what brain areas are they referring?
(Answer) I added the following statement to lines 47-50.
Food and smoking cues were both associated with increased blood oxygenation-dependent responses in the left amygdala, bilateral orbital cortex, and striatum. Brain regions identified here are involved in learning, memory and motivation, and their healing induction activity is an indicator of the simulative saliency of cues.
2. I do not understand what, “It is as per the teaching”, means. This is most likely a language issue but not one I can figure out. Regardless, using the identical phrase throughout the authors’ response is not appropriate.
(Answer) I understand that.
3. The confusion concerning the number of subjects remains (no changes to the text, addition of two arrows to the figure). They state in the abstract (line 15) that data from 134 people were examined, yet they state that 321 people were recruited for the study (lines 16 and 59). The authors’ response adds to the confusion as now they are talking about 36 classes and 36 levels.
(Answer) I changed lines 72-86 as follows.
We obtained data on 38 smokers (men 24 [16.1%], women 14 [8.1%]) and 283 non-smokers. And we conducted a food-dependent questionnaire survey on them in the second survey. We were unable to obtain data on 2 of smokers and 6 of non-smokers. We used the data from 36 of them (men 24, women 12) (the smoking group) because two declined to participate in the survey. We randomly extracted 36 people from 277 non-smokers, corresponding to smokers' condition (the sex, age: 20-39; 40-59; 60 years and over), annual income and educational background). Therefore, the number of data we used for comparison was 72 (smoking vs. non-smoking). A further 36 subjects (the non-smoking group) were randomly selected and matched according to sex, age (20 to 39 years old, 40 to 59 years old, 60 years old and over), annual income and educational background from among 277 non-smoking respondents (124 men and 153 women). In the second survey, we conducted detailed surveys on the 36 smoking subjects. Also, 31 of them (19 men and 12 women) were defined as being nicotine-dependent (nicotine dependence group), and we selected a further 31 non-smoking control subjects (no nicotine dependence group) from among the non-smoking group. Therefore, the number of data we used for comparison was 62 (nicotine dependence vs. no nicotine dependence).
4. The authors have adequately addressed my concern regarding the criteria used to classify subjects.
5. While the authors have addressed my issue with the phrase “except women” in their response, they have not modified the text to explain the phrase.
(Answer) I added lines 176-180 as follows.
Our study reported that smokers’ score was 6.2 ± 8.1 and non-smokers’ score was 2.3 ± 2.4 in food dependence (also nicotine-dependent’ score and non-nicotine-dependence’) in women. The difference between the two groups is greater than that of men. But, there were not statistically significant differences in these groups. The reason seems to be that the sample size was small.
6. The authors do not adequately address the major issue of using the mYFAS versus the full YFAS to determine food dependency other than to say they “would like” to use it in the future.
(Answer) I added lines 184-186 as follows.
Finally, mYFAS has been verified to be as reliable as YFAS. Thus, mYFAS has reduced the burden on participants by reducing the overall number of questions, enabling the assessment of food poisoning in large epidemiological studies [34, 35].
7. The authors have adequately addressed my concern regarding statistical analysis (bootstrap).
